# Assessment of the Effect of Cold Atmospheric Plasma (CAP) on the Hairtail (*Trichiurus lepturus*) Quality under Cold Storage Conditions

**DOI:** 10.3390/foods11223683

**Published:** 2022-11-17

**Authors:** Huiqian Xu, Wenhua Miao, Bin Zheng, Shanggui Deng, Shaimaa Hatab

**Affiliations:** 1Department of Food Science and Pharmaceutics, Zhejiang Ocean University, Zhoushan 316022, China; 2Faculty of Environmental Agricultural Science, Arish University, North Sinai 45516, Egypt; 3Faculty of Organic Agriculture, Heliopolis University, Cairo 2834, Egypt

**Keywords:** non-thermal, dielectric barrier discharge, hairtail, quality, shelf-life

## Abstract

Cold Atmospheric Plasma (CAP) is a novel non-thermal preservation method that extends the shelf-life of food. Therefore, this study investigated the effect of CAP on the quality parameters of hairtail (*Trichiurus lepturus*) during cold storage conditions (at 4 °C and RH range 45–55%). For that reason, different quality parameters including the total bacteria count (TBC), total volatile basic nitrogen (TVB-N), pH, thiobarbituric acid reacting substances value (TBARS), color, texture, and sensory evaluation have been measured. The hairtail was exposed to CAP at 50 kV voltage for 2, 3, 4, and 5 min. The results showed that the samples treated with CAP at 50 kV for 5 min had significantly lower (*p* < 0.05) TBC (7.04 ± 0.26 log CFU/g) compared with the control sample (8.69 ± 0.06 log CFU/g). Similar results were found concerning TVB-N, which strongly decreased in the treated samples (16.63 ± 0.03 mg N/100 g) in comparison with the control sample (22.79 ± 0.03 mg N/100 g). In addition, the CAP-treated samples had lower (*p* < 0.05) changes in color than those of the control group. With reference to the sensory evaluation, the shelf-life of CAP-treated samples (at 50 kV for 5 min) was longer than the untreated samples by about 6 days. These results led us to the conclusion that CAP can effectively delay spoilage and deterioration, slow the rise in pH, and maintain the sensory attributes of hairtail during cold storage conditions.

## 1. Introduction

Hairtail (*Trichiurus lepturus*) is one of the most economically important fish in China, where the catching rate is estimated at 10% to 20% of the total marine fish catch [1]. Hairtail is a highly nutritious fish, rich in unsaturated fatty acids including icosapentaenoic acid and docosahexaenoic acid, and in protein content as well. Therefore, protein degradation and lipid oxidation usually occur as a result of the endogenous enzymes and microorganisms’ activities during transportation and storage [2]. Like other seafood, hairtail is considered perishable food; therefore, the industry remains concerned about extending its shelf-life. Several preservation technologies such as electron beam irradiation, ultra-high-pressure sterilization, and chemical preservatives have been used to prolong the shelf life of the hairtail and maintain its quality [3,4,5,6].

Over the last few years, Cold Atmospheric Plasma (CAP) has emerged as a potential non-thermal technology that can modify the properties and structures of food, inactivate endogenous enzymes [7], and eliminate spoilage and pathogen microorganisms [8]. CAP is ionized gas generated by exposing the gas, either gas mixture or air, to remarkably high electrical field strengths that result in the formation of a wide range of reactive species including reactive oxygen species, reactive nitrogen species, hydrogen peroxide, excited oxygen and nitrogen, OH radicals, H_2_O^+^ and OH^–^ions [9,10]. Several studies indicated the antimicrobial activity of reactive oxygen and nitrogen species (RONS), which renders CAP a perfect surface decontamination method [11]. Therefore, CAP has been applied effectively to eliminate large groups of microorganisms, including bacteria, fungi, and even biofilms and spores [8]. In an earlier study, two log CFU/g reduction in the microbial load had been noticed in chicken breasts after CAP treatment at 100 kV for 5 min [11]. Similarly, the natural microflora of processed sushi effectively decreased after CAP treatment [10]. Fernández et al. [12] pointed out that CAP can kill Salmonella cells after a short time of treatment. In 2021, Koddy et al. found an improvement in the water-holding and color properties of CAP-treated hairtail, and the authors also reported remarkable inhibition in the activity of the crude protease extracted from hairtail fish after exposure to CAP for 240 s at 50 kV [13]. In comparison with traditional technologies, CAP possesses various advantages such as low-temperature treatment, short treatment time, inexpensive operation costs, no residue on the surface, high efficiency, and harmless technology (i.e., no hazardous chemicals and can provide nutritious and safe food), and it has environmentally friendly characteristics as well [14,15,16].

The available studies concerning the effects of CAP on the quality parameters of hairtail fish during cold storage conditions are quite limited. Therefore, the objectives of this study were (1) to evaluate the efficacy of CAP treatment at 50 kV for various treatment times in reducing the Total Bacteria Count (TBC) of hairtail, and (2) to measure the changes in the quality parameters including the Total Volatile Basic Nitrogen (TVB-N), pH, Thiobarbituric Acid Reacting Substances value (TBARS), color, texture, and sensory evaluation.

## 2. Materials and Methods

### 2.1. Chemicals

Analytical grade chemicals used in this research were obtained from Sinopharm Chemical Reagent Co., Ltd. (Shanghai, China).

### 2.2. Sample Collection and Preparation

Fresh hairtail (*Trichiurus lepturus*) fish were purchased from the local market in Zhoushan City, Zhejiang Province, China. The samples were transported to the laboratory in crushed ice (1:3, *w/w*) within 30 min. Immediately upon arrival, fish samples were washed with distilled water, the heads and tails were cut off, and the internal organs removed. The fish samples were cut into equal portions, then rinsed with distilled water, and the water on the surface was removed using sterilized gauze. The prepared samples, weighing about 0.4 kg, were packaged in commercial 270 µm-thick polyethylene trays (205 mm × 135 mm × 34 mm) and sealed using a packaging machine (MAP-H360; Suzhou Senrui Fresh-keeping Equipment Co., Ltd., Suzhou, China).

### 2.3. Plasma Treatment

A dielectric barrier discharge (DBD) plasma (Phenix BK130/3 AC Test Set 600 Series Processor, Phenix Technologies, Inc., Accident, MD, USA) was used to generate plasma, as mentioned by Tang et al. [7]. PP containers containing fish samples were placed between two parallel rounded aluminum plates with an outer surface of 155 mm and a distance of 75 mm between the two electrodes. The samples were subject to a voltage of 50 kV for various treatment times: 2, 3, 4, and 5 min, respectively. A group without CAP treatment was included as a control group. Afterward, the samples treated with CAP and the control samples were stored in a sealed condition at 4 °C for 15 days, and at days 0, 3, 6, 9, 12, and 15 the samples were analyzed for TBC and quality measurements.

### 2.4. Quality Analysis

#### 2.4.1. Determination of TBC

In order to quantify the effect of CAP treatment on the bacterial population of the cold-stored hairtail fish, the Total Bacterial Count (TBC) has been determined as an indicator according to the standard of GB/T 4789.2-2016. Briefly, a 25 g sample was added to 225 mL of normal saline and homogenized. The homogeneous were serially diluted using sterile normal saline. The pour plate method was used to enumerate the bacterial count on plate counting agar (PCA). Petri dishes were incubated at 30 °C for 72 h. All measurements were performed in triplicate and normal saline was included as a control.

#### 2.4.2. Determination of TVB-N and pH

Automatic Kjeldahl apparatus (KDN-520; Bangyi Precision Measuring Instrument (Shanghai) Co., Ltd., Shanghai, China) was used to evaluate Total Volatile Bases Nitrogen (TVB-N). A 10 g ground hairtail sample was soaked in 75 mL of distilled water for 30 min in a beaker. After homogenization, the samples were transferred to a distillation tube with 1 g of magnesium oxide. The titration was applied using 0.1000 mol/L HCl with the existence of methyl red as an indicator. The results were expressed as mg N/100 g muscle [17]. To analyze the changes in the pH values, a pH meter (PHS-3C; Shanghai INESA Scientific Instrument Co., Ltd., Shanghai, China) was used as described by Hatab et al. [18].

#### 2.4.3. Determination of TBARS

The method illustrated by Zheng et al. [19] has been used to determine the thiobarbituric acid reacting substances (TBARS). Briefly, 50 mL of trichloroacetic acid solution (7.5%, containing 0.1% EDTA-Na2) was added to 5 g of ground hairtail in a beaker and homogenized for 2 min, then centrifuged at 8000 rpm for 10 min at 4 °C. After filtration, 5 mL of supernatant was mixed with 5 mL of TBA solution (0.02 mol/L) and incubated for 40 min at 95 °C in a water bath. After cooling to room temperature, the absorbance values were measured at wavelengths of 532 nm and 600 nm, respectively, using a spectrophotometer (UV-5900 Ultraviolet-Visible Spectrophotometer; Shanghai METASH Instrument Co., Ltd., Shanghai, China). All measurements were taken in triplicate. 

#### 2.4.4. Color Analysis

The color difference in the hairtail sample with and without CAP treatments was analyzed during the storage period. Four different points at least of hairtail samples were taken for analysis using a colorimeter (CM-5; Konica Minolta, Japan). The formula given by Okutan et al. [20] was used to determine the total changes in the color of the sample (Δ*E*) value as follows:$$\Delta E=\sqrt{{\left(L_{1}-L_{2}\right)}^{2}+{\left(a_{1}-a_{2}\right)}^{2}+{\left(b_{1}-b_{2}\right)}^{2}}$$

#### 2.4.5. Texture Profile Analysis (TPA)

The food property tester (TAXT-plus; Stable Micro System, UK) was used to analyze the texture profile of hairtail samples (with a dimension of 50 mm × 50 mm) during the cold storage period [21]. The trigger force was set at 10 g and the samples were pressed to 30%. A P/6 flat-bottomed cylindrical probe was used, and the pre-test speed, mid-test speed, and post-test speed were all 0.50 mm/s.

#### 2.4.6. Sensory Evaluation

The odor, color, and organizational structure of hairtail samples during the storage period were evaluated by 40 trained panelists, according to Huss (1995) [22]. The average score of each item was considered as a result. The evaluation criteria are given in Table 1:

### 2.5. Statistical Analyses

The data were analyzed by IBM SPSS Statistics 19 statistical software, and the results were expressed as means ± standard deviation (significance level: *p* < 0.05). Each experiment was repeated three times (n = 3), and the mapping was performed by Origin 9.1 drawing software (Northampton, MA, USA) [7].

## 3. Results and Discussion

### 3.1. The Effect of CAP on the TBC

The microbial load is one of the most important factors leading to fish spoilage [23]. The effect of CAP on the microbial load of stored hairtail samples is shown in Figure 1. Throughout the storage period, all samples (both CAP-treated and untreated) had a significantly increased trend regarding the microbial load. The initial TBC of hairtail at 0 days ranged from 3.06 ± 0.08 to 2.22 ± 0.09 log (CFU/g) and reached 8.69 ± 0.06 log (CFU/g) in the control sample and 7.04 ± 0.26 log CFU/g in samples treated with CAP at 50 kV for 5 min, respectively, on day 15. It has been reported that the microbial load in the fresh fish usually ranged from 3.0 to 5.0 log (CFU/g) [24]. Our results indicate that CAP treatment could reduce the growth of microorganisms within a certain period. The reactive oxygen species (ROS), reactive nitrogen species (RNS), charged particles, and ultraviolet photons generated during the discharge process of CAP strongly inhibit or kill microorganisms’ growth on the surface of fish to a certain extent. [25]. In addition, the large number of ultraviolet rays generated during the plasma formation process could directly destroy the genetic material of the bacteria, and the active materials produced by the etching effect of the plasma could react with the proteins and nucleic acids in the bacteria and thus the bacteria loses its function [26,27]. By extending the treatment time, the number of active ingredients in the plasma and its interaction time with microorganisms increases significantly, and thus the degree of damage to bacteria increases as a result. It is well known that the lipid bilayer in the cell membrane is extremely sensitive to the electric field; therefore, after CAP treatment, the cell membrane is irreversibly destroyed [28].

### 3.2. The Effect of CAP on the TVB-N and pH

TVB-N is an unwanted compound produced upon protein decomposition by microbial or enzymatic activity [29]. Therefore, TVB-N is usually used as an effective index of the protein degradation level [30]. In this study, changes in hairtail samples increased generally during storage and ended up at 22.79 ± 0.03 mg N/100 g for the control sample on the 15th day of storage (Figure 2), while the CAP-treated sample at 50 kV for 5 min reached 16.63 ± 0.03 mg N/100 g on the same day. The increase in the TVB-N values explains the rise in pH value at the end of the storage period. The acceptable level of TVB-N in fresh hairtail should not exceed 30 mg N/100 g, according to Du et al. [31]. With this fact in mind, we can state that the CAP improves the shelf-life of hairtail samples compared to control samples. CAP strongly reduces the reproduction of bacteria, which leads to a decrease in the secretion of proteases by bacteria, and thus slows down the rate of amino acid deamination and reduces nitrogenous substances [32,33].

The changes in hairtail pH during cold preservation would reflect the freshness degree of fish. After catching the fish, a series of changes such as corpse stiffness and autolysis of fats and proteins occurs by the combined action of endogenous enzymes and microorganisms; these changes cause a shift in the normal pH value [34]. Generally speaking, the pH levels of control and CAP-treated groups declined at the beginning of the storage period and then increased significantly (*p* < 0.05) with the extension of the storage time (Figure 3). The observed decrease in the pH could be due to the anaerobic decomposition of glycogen and the degradation of ATP, which produces acidic substances such as lactic acid and phosphoric acid [35], while with extended storage time, the fish protein degrades into alkaline substances such as ammonia and trimethylamine by the action of endogenous enzymes and bacteria, which consequently increase the pH level [36]. Since the CAP could suppress the activity of endogenous enzymes as well as the bacteria, the changes in pH levels in the treated groups were, relatively, lower than in the control sample [37].

### 3.3. The Effect of CAP on the TBARS

Lipid oxidation leads to the formation of peroxides, which in turn break down into secondary oxidation products such as malondialdehyde (MDA) [11]. For this reason, MDA is commonly involved as a quality indicator to measure lipid oxidation and deterioration in fish and fish products. TBARS analysis reflects the interaction between two molecules of 2-thiobarbituric acid and one molecule of malondialdehyde (MDA) under the heated acidic condition, which results in forming a pink color that can be measured by a spectrophotometer [38,39]. The TBARS values of all treated groups showed a rising trend with the increase in the storage time (*p* < 0.05), as illustrated in Figure 4. On the 15th day of the cold storage, the TBARS value of the control sample and CAP-treated groups at 2, 3, 4, and 5 min were 0.66, 0.51, 0.38, 0.35, and 0.34 mg/kg, respectively. Hairtail is rich in unsaturated fatty acids; consequently, it is oxidized easily, which leads to growth in the value of TBARS [40,41]. The increased rate of TBARS values in the treatment groups was lower than that of the control group, and these findings are consistent with the results of TBC. In addition, it has been reported that the increase in TBARS values in treated samples could be due to the denaturation of fish protein during the CAP treatment, causing cell rupture and the release of strong oxidants such as free radicals and metal ions [42].

### 3.4. The Effect of CAP on the Color Difference

From the consumer perspective, fish color is one of the significant aspects. Therefore, color stability during storage is extremely critical for the fish industry sector. Table 2 and Table 3 present the changes in color and luminosity, respectively, of both CAP-treated hairtail fish and control samples during cold storage at g at 4 ± 1 °C. Results indicated that the color changes in the samples treated with CAP were notably (*p* < 0.05) lower than those of the control group. These findings revealed that the CAP treatment could maintain the color of the hairtail during cold storage at g at 4 ± 1 °C. While the color of fish is usually influenced by pigment levels and muscle structural properties, protein oxidation also leads to coloring loss in fish tissues [43]. Therefore, the differences in color value between the CAP-treated sample and control samples could be due to the destruction of endogenous enzymes including peroxidases during the CAP treatment, which slows down the oxidation of fish, thereby effectively maintaining the color [44]. Our results are in agreement with Mohamed et al. (2021), who reported significant stability in the lightness and whiteness of Nile tilapia fish subjected to dielectric barrier discharge–CAP (DBD–CAP) under 60 kV for 4 min [43]. However, in another study, no significant differences in color parameters were found between the plasma-treated chicken breasts and untreated samples [11]. The variations in other studies’ outcomes are probably correlated to the plasma chemistry associated with the type of discharge system used and its operating parameters.

### 3.5. The Effect of CAP on the Texture

The texture of fish is an important quality feature, and is affected by many factors such as the type of fish, the size, the fat and protein content of the fish, the microbial load, and the storage temperature and time [13]. The related parameters of texture are hardness, fracturability, adhesiveness, cohesiveness, springiness, gumminess, chewiness, resilience, etc. Four representative parameters, namely hardness, springiness, cohesiveness, and chewiness, were selected to analyze the texture characteristics of hairtail samples, both CAP-treated and untreated samples, during cold storage. As a general trend, all selected parameters showed a significant decrease (*p* < 0.05) throughout the storage period (Figure 5A–D). However, the degradation in texture for the samples treated with CAP at 50 kV for 5 min was lower than that of other treatment groups and the control group. According to their experimental results, Miao et al. [9] elaborate that the texture characteristics in the CAP-treated samples were higher than that of the control groups, and thus the CAP treatment could delay the deterioration of hairtail to a certain extent by inhibiting endogenous protease activities, which strongly affects texture changes [45].

### 3.6. The Effect of CAP on the Sensory Evaluation

Sensory evaluation is a fast, convenient, and effective tool implemented to evaluate and classify food quality [46]. The present study showed that sensory evaluation of control and CAP treatment groups of hairtail decreased significantly with storage time as given in Figure 6. As expected, at the beginning of the storage period all samples, either CAP-treated or untreated, were still fresh and had a natural smell, bright color, and tight muscle. Throughout the storage time and compared with the control group, all CAP-treated samples recorded a higher sensory score. On the last day of the storage period, the sensory evaluation for the samples treated by CAP at 50 kV for 5 min was three times higher than the control sample. These results indicate that there was a high correlation between physiochemical indicators (TVB-N, pH, and TBARS) and the sensory evaluation. CAP has bactericidal and antioxidant properties, which could inhibit the growth and reproduction of microorganisms, which in turn delays the degradation of the sensory parameters [47].

## 4. Conclusions

Our study reveals that CAP treatment at 50 kV for 5 min extended the shelf-life of the hairtail by an additional 6 days in terms of sensory evaluation. CAP-treated samples showed slower physiochemical changes and microbe growth rate during the storage period. It is noteworthy that the CAP treatment positively influences the quality parameters of hairtail fish in a treatment-time-dependent manner. It is concluded that CAP possesses great potential as a new technology to prolong the shelf life of hairtail. However, further research is needed to provide more understanding of the microbial inhibition mechanism of CAP and to assess the impact of reactive species.

## Figures and Tables

**Figure 1 foods-11-03683-f001:**
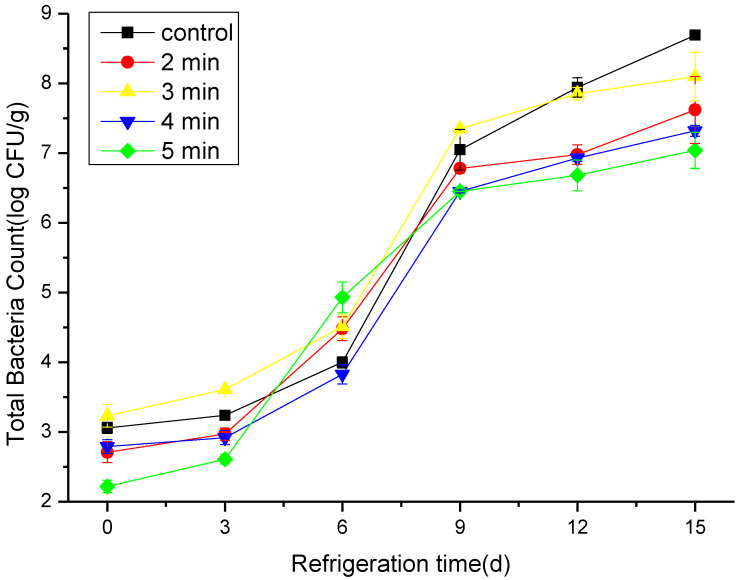
Changes in total bacteria count (TBC) populations in cold stored hairtail fish exposed to CAP at 50 kV voltage for various treatment times: 2, 3, 4, and 5 min. The non-treated sample is included as a control.

**Figure 2 foods-11-03683-f002:**
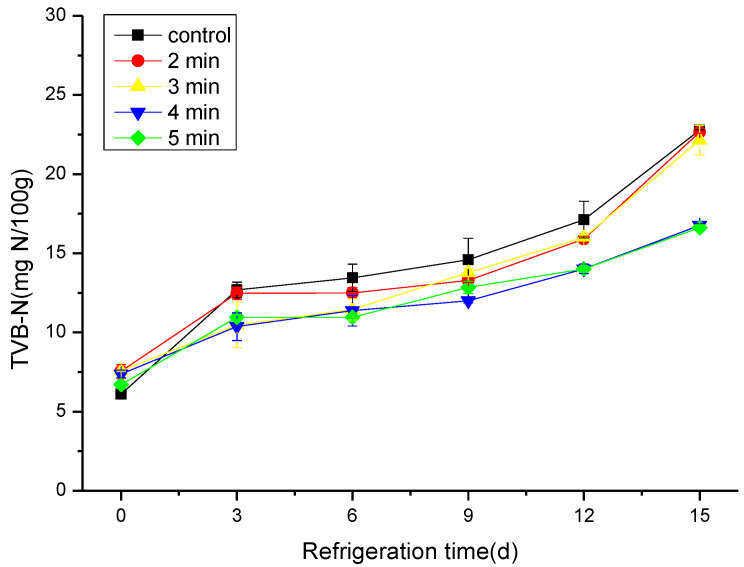
Changes in TVB-N of cold stored hairtail fish exposed to CAP at 50 kV voltage for various treatment times: 2, 3, 4, and 5 min. The non-treated sample was included as a control.

**Figure 3 foods-11-03683-f003:**
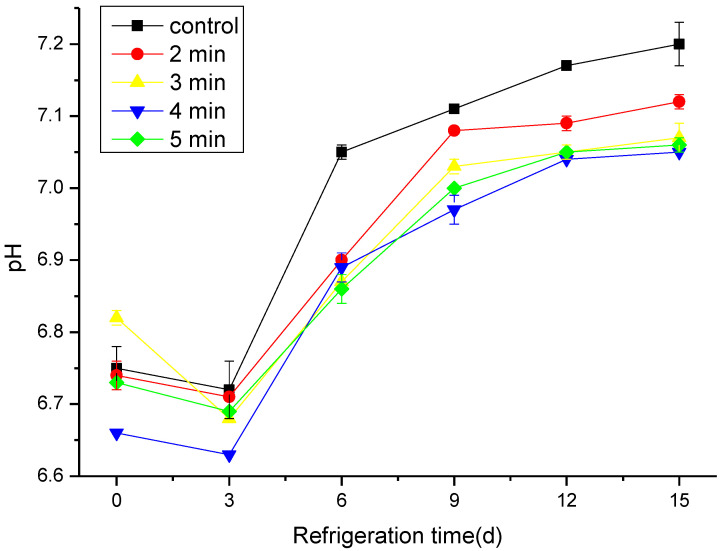
Changes in pH of cold stored hairtail fish exposed to CAP at 50 kV voltage for various treatment times: 2, 3, 4, and 5 min. The non-treated sample was included as a control.

**Figure 4 foods-11-03683-f004:**
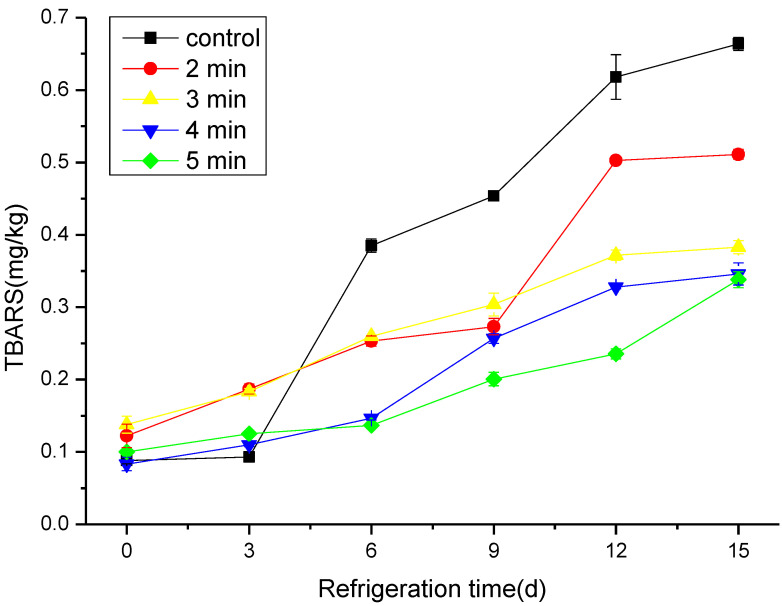
Changes in TBARS of cold stored hairtail fish exposed to CAP at 50 kV voltage for various treatment times: 2, 3, 4, and 5 min. The non-treated sample was included as a control.

**Figure 5 foods-11-03683-f005:**
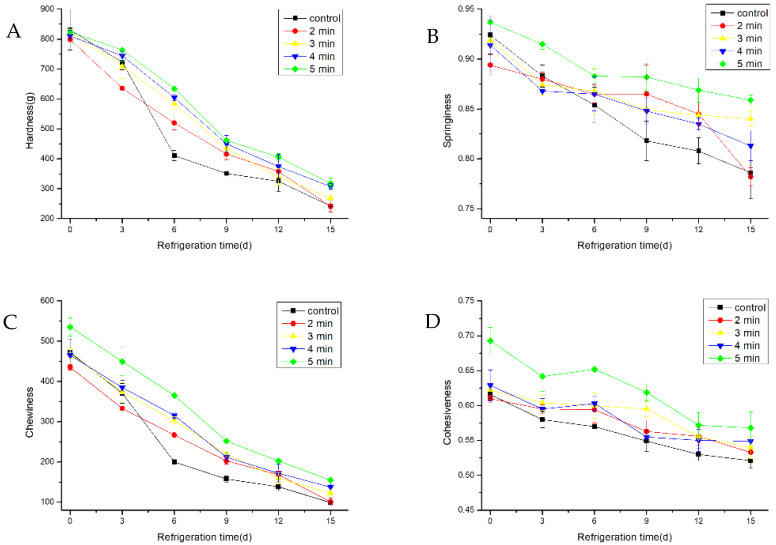
Changes in texture parameters: hardness (**A**), springiness (**B**), cohesiveness (**C**), and chewiness (**D**) of cold stored hairtail fish exposed to CAP at 50 kV voltage for various treatment times: 2, 3, 4, and 5 min. The non-treated sample was included as a control.

**Figure 6 foods-11-03683-f006:**
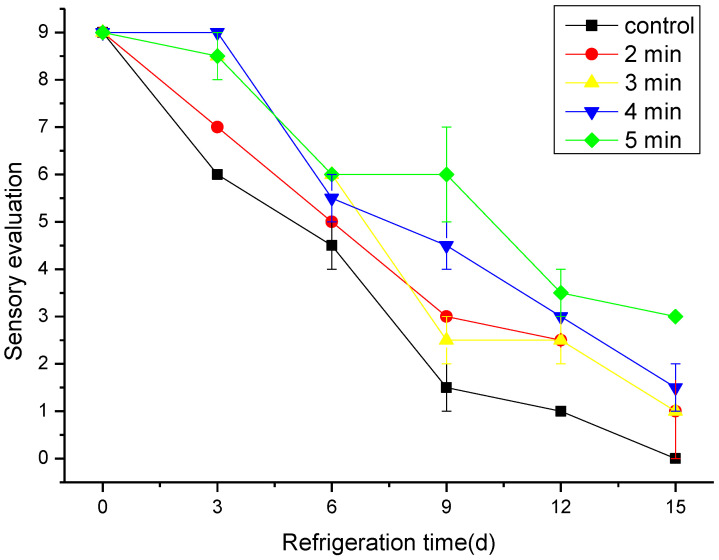
Sensory evaluation of cold stored hairtail fish exposed to CAP at 50 kV voltage for various treatment times: 2, 3, 4, and 5 min. The non-treated sample was included as a control.

**Table 1 foods-11-03683-t001:** Standards of sensory assessment.

	Best (3)	Good (2)	Poor (1)	Worst (0)
Odor	Inherent smell	Slight smell	Strong smell	Rancid smell
Color	Normally shiny	Dull shiny	Not lustrous	Discoloration
Texture	Normal, elastic	Slightly worse, elastic	Poor, less elastic	Worse, nonelastic

**Table 2 foods-11-03683-t002:** Effect of Cold Atmospheric Plasma treatment at 50 kV voltage on the variation of the total color difference of hairtail during cold storage conditions.

Refrigeration Time (d)	Treatment Time (min)				
	Control Group	2	3	4	5
3	3.22 ± 0.10 ^Ba^	2.57 ± 0.37 ^BCa^	1.47 ± 0.30 ^Cb^	1.20 ± 0.19 ^Cbc^	0.50 ± 0.07 ^Cc^
6	3.84 ± 0.38 ^Ba^	2.33 ± 0.22 ^Cb^	1.82 ± 0.24 ^BCb^	1.73 ± 0.47 ^BCbc^	0.72 ± 0.25 ^Cc^
9	5.50 ± 0.17 ^Aa^	3.40 ± 0.30 ^Bb^	2.56 ± 0.24 ^Bbc^	2.08 ± 0.34 ^ABCcd^	1.19 ± 0.37 ^BCd^
12	6.32 ± 0.17 ^Aa^	5.29 ± 0.14 ^Aa^	3.89 ± 0.37 ^Ab^	2.72 ± 0.48 ^ABbc^	2.05 ± 0.56 ^ABc^
15	6.34 ± 0.66 ^Aa^	5.48 ± 0.31 ^Aa^	4.18 ± 0.17 ^Abc^	3.28 ± 0.43 ^Abc^	2.46 ± 0.30 ^Ac^

Note: Capital letters indicate the significance of the difference between the data in the same column (*p* < 0.05); lowercase letters indicate the significance of the difference between the data in the same column (*p* < 0.05).

**Table 3 foods-11-03683-t003:** Effect of Cold Atmospheric Plasma treatment at 50 kV voltage for different treatment times on the luminosity of hairtail samples during cold storage conditions.

Refrigeration Time (d)	Treatment Time (min)				
	Control Group	2	Control Group	4	Control Group
0	88.95 ± 0.09 ^Ab^	88.95 ± 0.23 ^Ab^	88.86 ± 0.19 ^Ab^	90.61 ± 0.32 ^Aa^	89.54 ± 0.60 ^Ab^
3	86.52 ± 0.03 ^Bc^	86.41 ± 0.22 ^Bc^	88.17 ± 0.38 ^Ab^	89.68 ± 0.20 ^ABa^	89.11 ± 0.64 ^Aab^
6	85.30 ± 0.38 ^Cc^	86.74 ± 0.19 ^Bb^	87.05 ± 0.06 ^Bb^	89.02 ± 0.27 ^BCa^	89.34 ± 0.23 ^Aa^
9	83.46 ± 0.19 ^Dc^	85.67 ± 0.15 ^Cb^	86.53 ± 0.32 ^Bb^	88.69 ± 0.54 ^BCa^	89.38 ± 0.30 ^Aa^
12	82.71 ± 0.16 ^Dc^	83.71 ± 0.22 ^Dc^	85.51 ± 0.25 ^Cb^	87.96 ± 0.51 ^CDa^	88.44 ± 0.67 ^ABa^
15	82.62 ± 0.58 ^Dc^	83.47 ± 0.20 ^Dc^	84.78 ± 0.03 ^Cb^	87.43 ± 0.20 ^Da^	87.27 ± 0.58 ^Ba^

Note: Capital letters indicate the significance of the difference between the data in the same column (*p* < 0.05); lowercase letters indicate the significance of the difference between the data in the same column (*p* < 0.05).

## Data Availability

Data is contained within the article.

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
