# Peer review of "Assessment of the Effect of Cold Atmospheric Plasma (CAP) on the Hairtail (Trichiurus lepturus) Quality under Cold Storage Conditions"

_foods, 2022, doi:10.3390/foods11223683_

Round 1
Reviewer 1 Report
This manuscript describes interesting research. It contains some innovative data and contributes new information to the available body of scientific knowledge. Thematically, this manuscript is in the scope of the journal Foods and is scientifically valid. However, I believe that this text in certain regards needs to be decently edited and supplemented. My suggestions in this regard are as follows:
Lines 36 -51: Please add information about the number of similar studies conducted by other researchers and on the topic of Cold Atmospheric plasma in general. Please elaborate on their results, what are their main findings. I think that this part is too short and does not provide the reader with sufficient information to localize this paper in the landscape of previously conducted research.
Line 38: i.e. the sentence: Due to its non-thermal, economic, highly efficient....Could you provide the reference for this statement?
Lines: 62 and 63, i.e. text (packaging container of 205 mm*135 63mm*34 mm) please use the appropriate multiplying sign x.
Lines 210 -212: Please reverse the order of these two sentences.
All figures: Please make them all more original and clearer to read.
Please cite this paper's findings about the trends of publishing papers dealing with the application of cold atmospheric plasma applications for food treatments: Vukić M., Vujadinović D., Smiljanić M., Gojković–Cvjetković V. Atmospheric cold plasma technology for meat industry: A bibliometric review. Theory and practice of meat processing. 2022;7(3):177-184. https://doi.org/10.21323/2414-438X-2022-7-3-177-184
Author Response
Reviewer #1
Comment 1: Lines 36 -51-Please add information about the number of similar studies conducted by other researchers and on the topic of Cold Atmospheric plasma in general. Please elaborate on their results, what are their main findings. I think that this part is too short and does not provide the reader with sufficient information to localize this paper in the landscape of previously conducted research.
ü Response: The introduction section has been carefully revised and updated, more information about previous studies on CAP and its application in food system has been inserted. Line 40-60.
Comment 2: Line 38: i.e., the sentence: Due to its non-thermal, economic, highly efficient. Could you provide the reference for this statement?
Response: This sentence has been deleted and more information that serves our goals and objectives has been added with reference, Line 60.
Comment 3: Lines: 62 and 63, i.e. text (packaging container of 205 mm*135 63mm*34 mm) please use the appropriate multiplying sign x.
ü Response: This required modification has been done, Line 78.
Comment 4: Lines 210 -212: Please reverse the order of these two sentences.
- Response: The sentence has been revised and rephrased, more information and explanation for our results have been inserted and new references have been added. Line 225-242.
Comment 5: All figures: Please make them all more original and clearer to read.
- Response: All figures have been replaced with high clear version.
Comment 6: Please cite this paper's findings about the trends of publishing papers dealing with the application of cold atmospheric plasma applications for food treatments: Vukić M., Vujadinović D., Smiljanić M., Gojković–Cvjetković V. Atmospheric cold plasma technology for meat industry: A bibliometric review. Theory and practice of meat processing. 2022;7(3):177-184. https://doi.org/10.21323/2414-438X-2022-7-3-177-184
- Response: We could not download this research article thus it was hard for us to cite, however, we have cited more relevant and updated references

Reviewer 2 Report
The work on “Assessment of the effect of cold atmospheric plasma (CAP) on the Hairtail (Trichiurus lepturus) quality under cold storage conditions” is interesting and fits with the journal scope. The authors have applied the cutting edge technology for shelf life extension. The plasma activated water is more suitable than CAP to treat Hairtail. However, the author have applied only CAP. Please find my specific comments. I hope these comments will improve the scientific merit of this paper.
Abstract
The following important information’s are missing in the abstract
1. Provide the temperature and RH value maintained in cold storage
2. Complete the sentence “Our results indicated that the sample was treated with CAP for 5 min”
3. Provide the information on effect of CAP on texture and color
4. Add the statistical results in the abstract
5. The information’s on effect of CAP on the shelf life (how many days it has extended the shelf life) of hairtail is important
Keywords: Avoid the words used in the title
Introduction
Write the information on shelf life of Hairtail under atmospheric condition and cold storage condition
What are all the microorganism (mention the scientific names) affecting the shelf life of haritail
L40-42: Update the reference with the recent reference such as
“Impacts of cold plasma treatment on physicochemical, functional, bioactive, textural, and sensory attributes of food: A comprehensive review”
Write the novelty of this study before objectives
Materials and methods
L63: Mention the thickness of PP in micron
What is the reason behind the selection of 2-5 min exposure? Why the authors have not considered 1 min exposure?
The authors have optimized 5 min yield a good results. Then, why the author does not tried more than 5 min exposure?
TPA analysis: Mention the dimension of the sample used for TPA analysis
Results and discussion
Provide the original images of the samples taken during before treatment and during storage period
The authors have only highlighted the results. I recommend the authors to add more scientific reason/mechanism behind the results
Conclusion
The conclusion section is just the repetation of abstract. I recommend the authors to add a salient findings and future line of work
References
Please update the old references (published before 2016) with recent references
Author Response
Reviewer #2
ABSTRACT:
Comment 1: The following important information’s are missing in the abstract
- Provide the temperature and RH value maintained in cold storage.
- Response: The temperature and RH value have been added to the abstract. line 14.
- Complete the sentence “Our results indicated that the sample was treated with CAP for 5 min”
- Response: The sentence has been revised and rephrased to be more clear. line 18.
- Provide the information on effect of CAP on texture and color.
- Response: The required information has been added. line 22.
- Add the statistical results in the abstract
- Response: The statistical results have been inserted in the abstract. line 13-26.
- The information’s on effect of CAP on the shelf life (how many days it has extended the shelf life) of hairtail is important
- Response: The information’s on effect of CAP on the shelf life of fish by days have been inserted in the abstract. line 23-26.
- Keywords: Avoid the words used in the title
- Response: The keywords have been changed as recommended by reviewer. line 27.
INTRODUCTION:
Comment 2: Write the information on shelf life of Hairtail under atmospheric condition and cold storage condition
- Response: The introduction section has been carefully revised and updated, more information on shelf life of Hairtail under atmospheric condition and cold storage condition has been inserted. Line 39-61.
Comment 3: What are all the microorganism (mention the scientific names) affecting the shelf life of hairtail
- Response: It is hard to mention the scientific names of all microorganisms infected hairtail, and this is also beyond our objectives. We test the microbial load as quality indicator, our goal is not to do a microbiological study.
Comment 4: L40-42: Update the reference with the recent reference such as “Impacts of cold plasma treatment on physicochemical, functional, bioactive, textural, and sensory attributes of food: A comprehensive review”
- Response: All references have been updated and more recent references have been inserted
Comment 5: Write the novelty of this study before objectives
- Response: The novelty and necessity of performing such study has been inserted in the Introduction section. line 60-61.
MATERIALS AND METHODS
Comment 6: L63: Mention the thickness of PP in micron.
- Response: The thickness of used pp was 270 µm the information has been inserted. line 77.
Comment 7: What is the reason behind the selection of 2-5 min exposure? Why have the authors not considered 1 min exposure?
- Response: Based on our previous studies and experiments, we have noted that CAP treatment for 1 min does not have significant effect comparing with control, therefore this treatment has been excluded from our current study.
Comment 8: The authors have optimized 5 min yield a good result. Then, why the author does not try more than 5 min exposure? Response: We just think that economically wise, and from the quality perspective, it would be better to decrease the treatment time. In addition, in our previous studies we have tested longer time for different purposes its positive effect was not that significant compared with 5 min treatment.
Comment 9: TPA analysis: Mention the dimension of the sample used for TPA analysis
- Response: The dimension of the sample used for TPA analysis was 50 mm x 50 mm, we have included a supplementary data file which has some images illustrate the process better.
RESULTS AND DISCUSSION
Comment 10: Provide the original images of the samples taken during before treatment and during storage period
- Response: Original images of the samples taken during before treatment and during storage period have been provided in as a supplementary data file, because we were not planning to publish therefore the clarity of the images is not that good for publication but you can find then in the included.
Comment 11: The authors have only highlighted the results. I recommend the authors to add more scientific reason/mechanism behind the results
- Response: The results and discussion section has been revised and modified according to the comments we received from the reviewers. Many parts have been rewritten and more explanation has been inserted to enhance and improve this section. You can check the changes starting from line 141.
CONCLUSION
Comment 12: The conclusion section is just the reputation of abstract. I recommend the authors to add a salient findings and future line of work
- Response: This section has been rephrased and the future line of work has been inserted as recommended by reviewer.
REFERENCES
Comment 13: Please update the old references (published before 2016) with recent references
- Response: All reference sections have been updated and several old references have been replaced by new ones, however there are some others that could not be replaced.
